# Exploring the Needs of Informal Learners of Computational Skills: Probe-Based Elicitation for the Design of Self-Monitoring Interventions

Rimika Chaudhury*
School of Computing Science
Simon Fraser University

Taha Liaqat†
School of Computing Science
Simon Fraser University

Parmit K. Chilana‡
School of Computing Science
Simon Fraser University

## ABSTRACT

Informal learners of computational skills often find it difficult to self-direct their learning pursuits, which may be spread across different mediums and study sessions. Inspired by self-monitoring interventions from domains such as health and productivity, we investigate key requirements for helping informal learners better self-reflect on their learning experiences. We carried out two elicitation studies with paper-based and interactive probes to explore a range of manual, automatic, and semi-automatic design approaches for capturing and presenting a learner's data. We found that although automatically generated visual overviews of learning histories are initially promising for increasing awareness, learners prefer having controls to manipulate overviews through personally relevant filtering options to better reflect on their past, plan for future sessions, and communicate with others for feedback. Our findings have several implications for designing learner-centered self-monitoring interventions that can be both useful and engaging for informal learners.

**Index Terms:** Learner-centered Design—Informal learners—Elicitation Study—Self-monitoring techniques;

## 1 INTRODUCTION

Millions of people around the world are turning to informal learning resources online to develop computational skills [1, 4] and to keep up with the demands of remote work and learning [47, 65]. Informal learners can access a variety of educational content in different formats (e.g., articles, videos, forums, e-books) and pursue their learning at their own pace. However, these learners can face a number of barriers in their informal learning pursuits [9, 13, 68], particularly in self-monitoring their progress [11].

One of the key challenges that informal learners face in self-directing [40] their learning is that they often lack awareness of their own strategies and potentially unhelpful behaviors. For example, they may end up relying on suboptimal trial and error [11, 21] and oscillate between different media and resources without a systematic strategy. Furthermore, the feedback that a learner receives organically in a social setting, such as a classroom, is missing in informal learning settings where the onus is on the learner to monitor their own progress and assess comprehension.

Since activities of self-reflection have long been shown to be useful in formal classroom-based educational contexts [40], we wondered how these activities could be designed for informal learners who largely rely on online resources and pursue their learning individually. For example, *what if learners could observe their learning patterns across different media and study sessions? What if learners could monitor their time spent and reflect on their trial-and-error behaviors to better self-direct their efforts?*

*e-mail:rimika_chaudhury@sfu.ca

†e-mail: tliaqat@sfu.ca

‡e-mail: pchilana@cs.sfu.ca

In our research, we take a learner-centered design-oriented [63] approach to explore ways of allowing learners tap into their own learning experiences for self-reflection. To explore this design space, we take inspiration from prior work in self-monitoring and self-tracking in domains such as health, well-being, and productivity [15, 16, 19, 48], which has demonstrated several benefits of tracking progress using manual to automatic approaches. In the context of informal learning, recording and reflecting on learning activities could also be helpful for raising self-awareness at various stages of learning and learner's overall success [25, 44, 59]. However, recent research suggests that existing tracking tools offer little flexibility with data collection and presentation nor support scaffolding for goal-setting, which may limit the opportunities for user-driven self-reflection [14]. Thus, we explore the design space of self-monitoring interventions for informal learners, emphasizing goal-setting and reflection on progress.

In this paper, we use a two-part elicitation study to synthesize requirements for the design of self-monitoring tools and techniques for informal learners of computational skills. Using the design probes approach [67], we adapted features from existing tracking tools [15–17, 19, 39, 45, 48] to explore a range of ideas between two extremes: completely manual methods where the learner is deeply involved in both data collection and presentation [2, 24], and completely automated techniques where the learner's involvement is minimal [17, 38]. Most of the ideas that we explored leverages *semi-automated* ways of self-monitoring [15].

In our first study, we showed 8 participants our design probes in the form of paper-based mock-ups to elicit their perceptions of what kind of data is useful to reflect on, and understand the extent to which learners may want to be involved in self-monitoring activities. We learned that visual overviews of learning histories can be insightful for learners and provide them with a way to meaningfully engage with their data. However, we were still unclear about how learners may use any of this information for reflecting on their learning and goal-setting as our paper-based mock-ups did not capture the dynamics of interaction for reflective activities and planning.

In our second study, we used the insights from our first study to design three interactive probes that varied the presentation of the overviews along three dimensions: temporal, resource-type, and topic. Our second elicitation study with 12 participants focused on understanding how learners would use the different interactive overviews to evaluate past efforts and plan the next steps. We found that semi-automatic approaches appealed to our participants the most, consistent with other studies [15]. Participants appreciated the at-a-glance summary provided by the broad categorizations of resources, and welcomed the granular breakdown of daily activities with an integrated to-do list (e.g., in the Temporal Overview) to reflect on multiple goal-pursuits. Most participants felt hesitant about the completely automatic generation of subtopic clusters in the Topic-based Overview for the lack of transparency and control it offered. Participants expressed eagerness to be involved in actions (e.g., tagging, annotating) that could improve the utility of the overviews for planning and sharing.

The key results from both of our studies together provide an initial set of requirements for designing self-monitoring interventions for

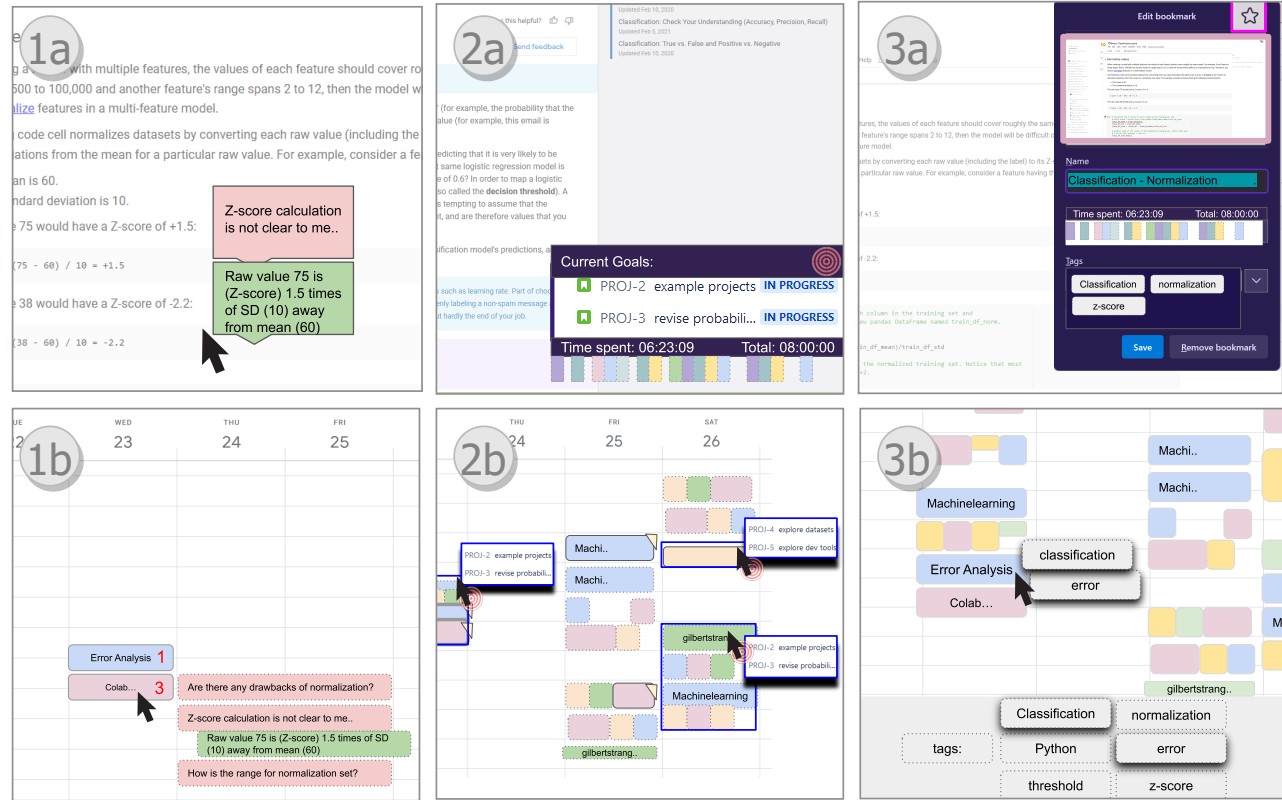

Figure 1: Three examples of paper-based visual overviews used in Study 1. Design idea #1: Muddy point approach is illustrated by Figure 1a. and 1b. Design idea #2 Goal-setting is shown in 2a. and 2b. The third column shows design idea #3 Vocabulary based filter.

informal learners. We make several design recommendations for promoting self-reflections to help learners leverage their learning histories, and be more engaged and in control of their informal learning pursuits. In this paper, we make the following contributions:

1. Insights into the kinds of data learners consider meaningful for reflecting on their learning processes;

2. Initial low-fidelity and high-fidelity design probes that helped us gain insights into learners' perspectives on how data could be used for self-reflection and planning for future learning;

3. Implications for designing self-monitoring tools for informal learners that provide a balance between manual and automated tracking methods;

## 2 RELATED WORK

Our work builds upon research on informal and self-directed learning of computation skills, techniques used in self-monitoring interventions in HCI, and data-driven approaches for supporting learning.

### 2.1 Informal and Self-directed Learning

Recent studies in HCI have brought to the forefront, the growing population of informal learners that is interested in learning programming and other advanced computational skills [11, 13, 29, 30, 68]. These studies have revealed that informal learners can face challenges due to different factors, such as the underlying programming environments [41], quality of lessons [29], and the learners' own assumptions and biases [11, 57]. A consistent theme that has been observed is that informal learners often engage in exploration and trial-and-error strategies during resource selection and implementation [11, 20, 21] and lack opportunities for reflective and critical

thinking [46]. Although recent efforts are starting to design interventions for self-reflective practices in formal programming courses [42–44, 58, 59], there is limited understanding of how and whether these interventions could work in informal and less structured environments where learning is more sporadic and spread across various mediums [11]. We extend the literature by exploring how self-monitoring interventions could be designed to support informal learners and the types of data and interactions that these learners would find useful.

### 2.2 Self-Monitoring Interventions in HCI

Prior research on self-monitoring interventions in areas such as health, well-being, and productivity, has shown that self-tracking practices can promote self-awareness and self-reflection [19, 35, 37]. However, researchers have noted that the burden of manual data collection could discourage people from engaging with the practice on a long-term basis [15, 24, 45]. Attempts to tackle the challenge through automatic data capture techniques has revealed that lack of engagement with data collection significantly reduces users' sense of awareness, accountability, and involvement [15]. At the same time, the rise of non-digitized, manual styles of tracking has revealed how slow, physicalized and deliberate involvement with personal data may promote self-reflection and self-expression [2]. In response, researchers have started exploring semi-automated approaches that combine automatic and manual approaches to collect data, and have called to attention the need for understanding the users' motivation and context to maximize the benefits of each approach [15, 17, 39, 45]. In addition, simply viewing tracked data may be insufficient for insight and behavior change [14]. Persuasive strategies such as reminders, suggestion, incentives, and social roles have been combined with self-monitoring techniques to encourage behavior

Table 1: Four design attributes that were considered in the design of the paper-based mock-ups, based on the literature of self-tracking tool designs in productivity, and health and well-being domains.

**1. Data recording:** Automatic tracking through sensors or logs, may reduce the burden of capture [3, 17], whereas manual tracking allows users to be more aware and engaged with data [5]. Leveraging both forms of data-collection through semi-automated approaches provides the flexibility of shifting the control between the user and the system as appropriate [15]

**2. Data presentation:** The layout of the data should allow users to glean insights through exploration, and therefore should provide glanceable summaries, as well as allow the users to manipulate the visualization to discover details on demand. Such goals can be supported through interactions like selection, filtering, and zooming [23, 70]

**3. Motivational components:** Self-monitoring tools are often designed with behavior change as a target outcome. Such goals are often supported by persuasive techniques such as reminders, nudges and prompts, for goal-setting and goal-adherence [32, 49].

**4. Material considerations:** Physical materiality of tracking tools often allow for mindful, slow-paced explorations and self-reflection [2, 66]. On the other hand, digital tools can afford long term tracking, ease data collection overhead, and offer powerful interactions for exploring data [55].

change [18, 52]. We take inspiration from these interventions to elicit requirements for helping informal learners self-monitor their progress and assess their perceptions of different data gathering and presentation techniques.

## 2.3 Data-Driven Approaches for Supporting Learning Reflections

Researchers in Education and Learning Sciences are increasingly looking at data-driven approaches to scale online learning and offer users more control of their learning. Common approaches to support self-direction [56] involve encouraging goal-setting through selection of skills, self-evaluation, and reflections through feedback on assessments [8, 31, 61, 69]. Increasingly, learning analytics are being explored to leverage key learning events (e.g., views, quizzes, discussion comments) for feedback. However, such events are confined within the same course or learning platform, and the feedback are analyses of student performance, based on fixed metrics, intended primarily to inform the instructors and the administrators of learner engagement [7, 26, 34]. While learning analytics studies have addressed several aspects such as, the technical challenges of implementation [7, 26], domain-specific challenges in learning (e.g., computer science, medicine) [22, 44, 59], as well as design challenges [26, 28], they have rarely considered the more sporadic nature of informal learning spread across different resources. Additionally, these prior works provide limited insight into how learners perceive and utilize such tools, extend little information of the design process and considerations, and tend to overlook the importance of offering learners ways of interacting with the data that arises from their learning events, for self-exploration and discovery. We distinguish our work by taking a learner-centered, iterative design approach to first understand from the learners' perspective what kind of data [69] and interactions might be useful for reflections.

## 3 STUDY 1: ELICITING REQUIREMENTS USING PAPER-BASED MOCKUPS

Our research is guided by learner-centered design [63], which recognizes the diversity in learners' objectives, motivations, and challenges. We aimed to understand informal learners' perceptions of self-monitoring techniques and identify key considerations for designing self-directed learning tools. To achieve this, we utilized a qualitative design-probe approach and adapted self-monitoring intervention attributes from other domains, such as health and productivity (see Table 1). In Study 1, we assessed learners' perceptions of the strengths and weaknesses of manual, automatic and semi-automatic approaches of self-monitoring techniques in the context of informal learning. Our key research questions were: *What kind of data would informal learners find useful for self-reflecting on their learning? To what extent are learners willing to be involved in data collection and presentation tasks for self-monitoring interventions?* We explored design interventions using paper-based mockups (simplified sketches of visual overviews) as they allowed us to illustrate a wider range of ideas and prompt participants to speculate on the utility of seeing of their personal data and learning patterns.

### 3.1 Exploring Visual Overviews using Paper Mockups

To prepare the 6 paper-based mock-ups, we considered the methods of data recording, data presentation, the presence of motivational components such as goal-setting and materials for prototyping (Table 1). To explore the spectrum of automatic to manual forms of recording data, we included probes with automated tracking with limited user-control (design idea #3), as well as designs with manual recording methods (design idea #6). In terms of presentation, we used a weekly summary layout, with daily and hourly details [50, 53] and adapted them based on the learning scenarios. We also considered the use of pre-session goal-setting as well as in-session reflective prompts for goal-alignment (design idea #2). We probed about material preferences by using mockups that implied physical (design idea #6) tracking as well as digital tracking (design ideas #1 - #5).

**Design idea #1 - Muddy Point:** Learning sciences research shows that quizzing and self-explanations [12] support knowledge retention and comprehension, even when the questions are learner generated [33]. This design explores the idea of externalizing and visualizing resolved and unresolved questions to facilitate reflections on muddy-points. The probe mocks up the interaction for annotating questions and answers within resources, and presents this information in a calendar-based overview, alongside the relevant resource (see Fig. 1.1a and 1.1b).

**Design idea #2 - Goal Setting:** Learners may have different goal-orientations, such as to master a skill, improve performance, or avoid failure or a combination of these [51]. This design explores how learners might define and track their goals through a Kanban [50] inspired board, commonly used for software management. The probe introduces the idea of staying goal-oriented by mocking up pop-up interactions for manually associating priorly set goals with newly visited web-pages while studying (see Fig. 1.2a). The overview shows participants how resources could be grouped based on goal-associations to facilitate reflections on goal alignment at the end of the learning session (see Fig. 1.2b).

**Design idea #3 - Vocabulary-based Filter:** Since learners initially have a limited vocabulary in a new learning domain [27], we explored the idea of automatic generation of keywords based on content viewed and saved by learners (see Fig. 1.3a). The design presents a weekly collection of keywords that can be used as filters to scan for related saved resources in the overview itself (see Fig. 1.3b). This idea aimed to gather insights on completely automated support for collecting data regarding newly learned vocabulary, and how they could be used for reflecting on resources.

**Design idea #4 - Predicting Usefulness:** Informal learners often experience difficulty and uncertainties in making decisions regarding resources [54]. This may be a result of lower competency as learners familiarize themselves to new content. However, reflecting on the utility of a given resource may support learners in making more thoughtful choices [10]. This design mocks up interactions to probe into how our participants' perceived making explicit judgements about visited webpages to reflect on their usefulness. The overview displays a simple browser-based form to insert notes, in-

dicate emotions about the resource, view a count-down timer and indicate useful subsections within resources, through check-mark annotations.

**Design idea #5 - Cross-Referencing:** Informal learners often tend to assimilate information from multiple sources. Since establishing connections between diversified sources facilitates comprehension and reflection [62], this design explores the idea of cross-referencing as an interaction while browsing, and a potential learning-activity data to be visualized. The mock-up shows a webpage where a link to an external page, that is bookmarked in a prior session, can be added. The calendar-based overview reflects which resources contains cross-references. It also includes a chatbot for automatic analysis of visited webpages to gather insights on what learners feel about automatic summaries of their browsing activities.

**Design idea #6 - Bullet Journals**: This mock-up elicits perceptions of manual and physical ways of tracking learning by presenting data using two pages of a pocket-sized diary. While the journal itself offers some degree of openness, the small size and bullet-point approach [6] creates a bounded-ness. The data in the mock-up reflects dates, resource titles, and a comment or a question logged by the learner in the scenario we were using. The design allows for flexibility through the use of symbols to add more meaning to the bullet point entries, such as question marks to indicate points of confusion or difficulty, single arrows and double arrows to indicate the idea of scheduling tasks for a future date.

## 3.2 Study Procedure and Analysis

Participants first filled out a brief questionnaire that included demographic questions, their personal definition of progress in a learning context, and success in that context. We used each paper-based mockup as a conversation starter and introduced participants to an informal learning scenario to help them understand the context for each mockup. We next asked the participants to judge the pros and cons of each idea based on its capacity to facilitate reflections on their learning, including judgment of quality of resources viewed, time spent, and extent of progress. The interviews were conducted in person and each session lasted approximately an hour. All participants were offered CAD 20 gift-cards for their participation.

**Participants:** Our goal was to recruit participants who were learning complex technical skills using informal online resources. We reached out to the contacts of the research team and to others using snowball sampling, and through the university mailing list. Eight participants (4M/4F) who were all students (6 graduate and 2 undergraduate students) between 19-35 years of age signed up for our study. Our participants reported self-learning technical skills such as machine learning (ML), web development and data visualization.

**Analysis:** The interviews were audio recorded with the participants' consent and later transcribed. Two researchers were involved with analyzing the qualitative data from the interviews. We used an inductive analysis approach [64], beginning with open coding to inspect each transcript. While coding, we considered how the responses were related to participants' perception of meaningful data for self-reflections, and their perceptions on data collection methods. We assigned multiple codes where necessary and had regular discussions with the research team to reconcile our final coding scheme. We performed axial coding to explore themes around our research questions and synthesized key insights as our results.

## 3.3 Key Findings from Paper-Based Elicitation

All participants found the idea of self-monitoring to be useful for gauging learning progress and leaned towards semi-automatic approaches of tracking. All but one (7/8) participant explained that while automatic methods could provide a convenient and systematic way to record their learning attempts, having manual control over certain aspects of data collection and presentation could improve the utility of visual overviews. Most participants (6/8) wanted to be able to indicate to the system what kind of information to record, such as the frequency of concepts marked as relevant, difficult, urgent or important. Additionally, participants wanted to be able to add, remove or edit data in the overviews to make them more accurate, specific or useful for planning and prioritizing. Lastly, participants shared some concerns regarding goal-setting and gauging progress.

### 3.3.1 Overviews create awareness of learning processes

Participants expressed that visual overviews could help them identify where they were spending time, and assess the extent of their progress during the period. Most of the participants (6/8) found the hourly breakdown of daily activities as a useful indicator of productivity and performance. The participants shared that the automatically generated overviews (see Fig. 1.1b, 1.2b, 1.3b) could additionally serve as a reminder of the topics within resources that they had viewed, and tasks they had accomplished during the week. Participants (7/8) acknowledged an overview could facilitate scanning and searching for resources they found useful, and could help them save time while resuming a subsequent study session that requires revisiting the same resources. Half of the participants (4/8) were in the habit of collecting useful links on note-taking apps for future access and mentioned that the overviews could serve as gateways to their personally curated collections of resources. P04 added that easier recognition of relevant resources could also facilitate recollection of helpful strategies for a future task: *"...because you are clearly grouping everything, it's easier to look back at later.. I do open up old projects if I am trying to remember how I did something before."*

While all participants agreed that the time-based overviews could help them filter information based on days and hours, they also imagined other ways to parse their learning histories. Three participants said that they cared less about the exact time spent on a resource or activity and were more interested in gaining an overall idea about their engagement. Most participants (6/8) were interested in knowing how many novel concepts they had learned during a given study session (see Fig. 1.3b), or gleaning the topics they had been studying during the specified time (one week, in our study).

### 3.3.2 Semi-automatic data-recording for purposeful revisits

The majority of participants (7/8) indicated that they were skeptical of the automatic identification of topics from the visited links, and preferred to have some control over the topics presented in the overviews. While participants preferred minimal engagement with data collection processes during the study session, they were willing to fine-tune the automatically detected topics to better indicate on the overviews which concepts they had studied and found relevant.

Many participants (5/8) were apprehensive about seeing every visited link (and search times) on the overview, as much of this effort could have been wasteful. They wanted the overviews to mostly serve as a way to look back on fruitful pursuits, and only wished to see resources with which they had meaningful interactions. These could be resources that they have accessed repeatedly, or annotated with highlights, comments and questions, or saved. In P01's view: *"[Progress] is not about the time we spend.. it is about having some sort of questions in mind. If [...] the questions are answered, then I made progress."*

Almost all participants (7/8) mentioned that overviews should present learning activities that could improve the value of a resource. As examples, participants mentioned activities like classifying resources into broad categories based on topics or content type (e.g., segregating code tutorials and conceptual content), identifying relevant segments within articles or videos, and associating tasks with resources. Participants also saw value in using visual overviews to indicate level of perceived difficulty and relevance of resources to

current learning interests, and prioritize resources. P04 noted that despite being annotated and deemed useful by the learner, resources may still end up being difficult to revisit: *"you need to take action on them later in order for [annotations] to have value"*. In the next subsection, we touch on the potential of using overviews for helping learners evaluate progress and identify actionable items.

### 3.3.3 In-context to-do lists for gauging learning progress

The majority of the participants (7/8) mentioned specific ways in which visual overviews could help them achieve a sense of progress. For example, five participants mentioned how a to-do list showing an account of completed and pending tasks and the provision to strike out tasks could help them assess their progress. However, four participants expressed concerns about using goal-oriented interventions (design idea #4) stating that initial goals may be *"super vague or sometimes incorrect"*(P07) and tend to evolve over time.

Participants expressed some hesitation in writing down goals before a learning session, as they usually figured it out "on the go". P03 shared how he tended to submit his code in *"one large commit on GitHub"* instead of thinking about it in chunks. However, he acknowledged that grouping resources based on the learners' areas of interest or projects could be helpful (see Fig. 1.2):

> It's almost like you're giving this resource a metric, because you've already given a metric in your head, you come into this resource with a bias, which is a good thing...you're able to reflect on yourself and on this resource, whether it's useful, or not. (P03)

P06 also shared how the overviews could provide an objective measure of progress in terms of the number of novel topics covered (see Fig. 1.3), and could be useful for sharing for feedback as it is *"relatable to others"*. For more personal measures of progress, participants indicated that they were interested in seeing the "extent" of progress in the direction of a larger goal (e.g., a summary of consumed content, or an account of answered questions). P05 mentioned that partial progress meant completing *"75% of the web page"* or trying out *"example codes from tutorials and copy[ing] line by line [to] see what happens."*

From our paper-based study, we learned that participants wanted to improve performance at certain repetitive tasks (e.g. refining output of a model), or determine if they had been successful in adapting existing examples to their specific needs. However, the paper-based mockups were limited in that they did not capture the dynamics of possible interactions learners may want to carry out while using overviews for reflections and planning. We wanted to probe more into what would make reflections more engaging and useful, while using different types of visual interactive overviews.

## 4 STUDY 2: EXPLORING THE DESIGNS OF INTERACTIVE OVERVIEWS

In Study 2, we refined the requirements from Study 1, by investigating how learners perceive interactive visual overviews as means of reflecting on their learning patterns, and gauging progress. We designed three interactive probes (see Fig. 2, Fig. 3, and Fig. 4) that highlighted different aspects, including time spent, resource types used, and topics covered in a learning session. As *at-a-glance summaries* are shown to be useful for providing quick insights on progress [23], we varied the data presentation to provide learners with different types of insights into their learning experiences. Our research question was: *How do learners make use of interactive learning overviews that vary in presentation format (e.g., time-based, topic-based, resource-based) to reflect on recent learning efforts and to plan the next steps?* Next, we describe the design of the three interactive probes, and the motivations behind each design.

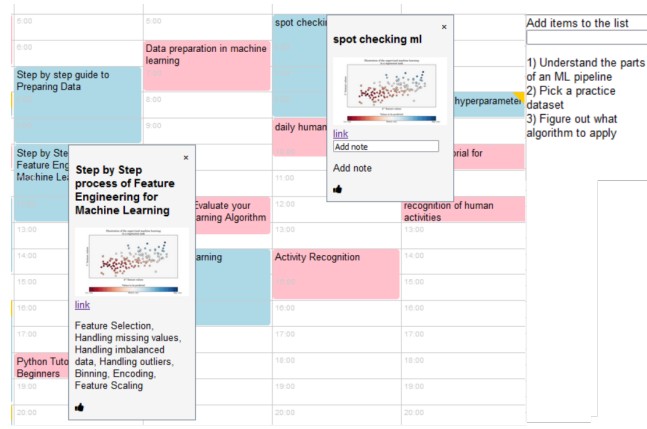

Figure 2: *Temporal Overview* uses a calendar-based time-boxed layout to represent visited and annotated (red boxes) resources, with details of each resource displayed on a card, on demand. A basic to-do list is included.

### 4.1 Temporal Overview

**Motivation:** Our participants perceived the calendar-based overviews of learning resources to be useful for developing an awareness of productivity, and have the potential for optimizing the time to look up saved resources. We wanted to further disentangle the benefits and drawbacks of temporal overviews.

**Description:** To help users track their learning and prompt reflection, the Temporal Overview (see Fig. 2) uses timeboxes [53] to show a weekly spread of the duration (hours) spent on each resource. Timeboxes are colored red when annotated, or blue when unannotated. The timeboxes display a yellow dog-ear bookmark when a resource is judged and marked as important by the learner. We included keyword-base filtering based on the 9 most relevant keywords from the contents of the selected webpages, with the goal of encouraging reflection of newly learned concepts. Clicking on timeboxes displays resource details, such as snapshots, notes, and usefulness judgments (thumbs-up icon). This overview includes a basic in-context to-do list.

### 4.2 Resource-based Overview

**Motivation:** Our first study revealed that helping learners recognize different types of resources, such as segregating code tutorials from conceptual content, could prompt reflection. In addition, we wanted to investigate whether providing information about the sequence of resource access could also encourage reflective thinking.

**Description:** The Resource-based Overview (see Fig. 3) displays the learning medium (e.g., videos, articles, tutorials, forums, publications) and encourages reflection on content preferences. We wanted to probe the importance of resource titles for interpreting overviews and ways to simplify time-spent information. Resource circles, varying in size and displaying a logo of the source, represent each resource, with the title available on-demand through mouse-hover. Three circle sizes represent relative time-spent per resource. Arrows indicate the sequence of resource access, with darker shades of blue indicating higher frequency of movement. Red borders indicate annotated or questioned resources. Details are available on-demand using a details-card per resource approach. This overview also includes an in-context to-do list, similar to the Temporal Overview.

### 4.3 Topic-based Overview

**Motivation:** Based on Study 1 findings, we investigated the potential of topic or keyword-based word clusters in promoting self-reflection.

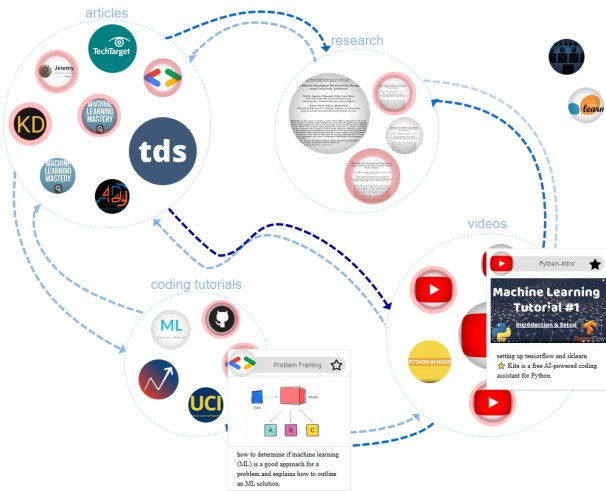

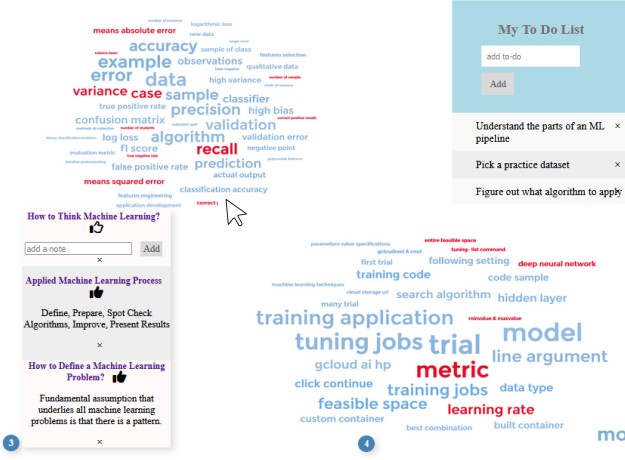

Figure 3: An example *Resource-based Overview* that represents resources grouped by resource-type (e.g., videos, tutorials). The arrows represent cumulative movements from one resource-type to another, with more frequent movements shown in a darker shade. To-do list is not shown in this figure.

Figure 4: *Topic-based Overview* clusters resources based on similarity of topics; word clusters show relevant keywords from each collection. Collection of resources are available in a list, on-demand.

Participants indicated that an account of newly learned content, such as keywords and concepts, could provide a sense of progress.

**Description:** The Topic-based Overview (see Fig. 4) presents word clusters that coalesce relevant keywords found within saved or annotated web-pages based on the similarity of topics. For example, resources related to different topics, such as algorithms, introductory statistical concepts, advanced training and tuning related concepts, and toolkits, are separated out into different clusters. However, the criteria for such grouping are left to the participants to interpret from the words in each cluster. Red highlighted words suggest outstanding questions related to the corresponding topic. Word size implies the relative time spent on the underlying resource and topic. The source webpages for each cluster are available on demand as a list, with personal notes and judgments of usefulness. Consistent with the other probes, the design includes an in-context to-do list.

### 4.4 Implementation of Interactive Probes

All of our interactive probes were semi-automatic by design as they showed the potential to capture certain data automatically (e.g., the time of visit, titles, snapshots, and resource URLs), but included opportunities for users to manually provide additional details (e.g., assessment of resource usefulness, annotations, notes, and to-do items). Each overview also included some capacity for edits (e.g., removing or adding notes and resources) which we used to explore perspectives on editing for personal use versus sharing with one other person for feedback. We used the same set of mocked-up data and shared characteristics to design each visual overview. The overviews summarized the past week, showed learner interactions with resources (e.g., annotated questions, bookmarks), and included a to-do list. Basic interactions were available, such as filtering, selecting items, and navigating to external links. The in-context to-do list was designed to be simple, encouraging participants to share their perspectives on additional useful actions for reflection and planning. The probes were implemented as high fidelity prototypes using JavaScript frameworks and libraries (React and JQuery), along with wireframing toolkits (Axure RP).

### 4.5 Study Procedure and Analysis

For our second elicitation study, we conducted semi-structured interviews using interactive probes to facilitate discussion. We first introduced participants to a scenario where a learner, Jay, uses online informal resources online to learn ML. Jay's learning activities online (e.g., visited websites, annotations, time) were illustrated in our probes as visual overviews. The data used to create the overviews were mocked-up by the researchers. We asked our participants to assume that such overviews could be generated automatically in real-time as they proceeded through their learning, and reminded them that the overviews were only partial representations of the learning activities for one week. While the general structure of the interview questions remained the same as Study 1, we refined certain questions to probe more deeply about specific design attributes using the interactive probes. For example, we asked our participants to interact with each probe and share with us how they may use such visual overviews to reflect on their past learning sessions, plan ahead for future sessions, and how they perceived the pros and cons of each presentation style. We also asked them for their thoughts on how they may use such overviews to ask for further feedback. The interviews were conducted in person when possible. When a participant requested for a remote session, it was carried out over Zoom where the researcher shared the probes through screen-share and audio-recorded the interview. Each session lasted approximately an hour and all participants were offered CAD 20 gift-cards.

**Participants:** We recruited 12 new participants, P09 to P20, (8F/4M) through personal contacts of the researchers and through word-of-mouth. Our participants' ages ranged from 18-44, and they were a mix of university students (undergraduate and graduate) as well as professionals from the software industry, engineering and art, with several years of experience. The 4 students came from CS and Biology departments. Each participant had experience learning technical skills informally, during their careers.

**Data Analysis:** The interviews were audio recorded with the participants' consent and later transcribed. Two researchers analyzed the qualitative data from the interviews using an inductive analysis approach, similar to the first study. We began with open coding [64] as transcripts became available, and iterated with the team to arrive at a final coding scheme. While coding, we considered participants' perceptions about more nuanced aspects related to reflections and planning, using temporal, topic based and resource-based presentation in the three overviews. Following this, we performed axial coding and diagramming to explore themes around our research question. The key insights are synthesized into 3 main themes around how learners perceived the interactive overviews.

## 4.6 Learners' Perceptions of the Interactive Overviews

Overall, participants expressed that the Temporal Overview and Resource-based Overview were "visually pleasing" and easier to interpret. Participants usually began by scanning for the overarching topics followed by sub-topics. Next, they focused their attention on their own knowledge and areas of weakness, or pending action items. Finally, participants desired the ability to edit the overviews to make them more suitable for seeking or offering feedback.

### 4.6.1 Useful to evaluate the quality of time spent learning

Participants considered the weekly overviews to be helpful in serving as a reminder of the recent learning activities. Most participants (8/12) ranked the Temporal Overview (see Fig. 2) as their first preference as it gave them a quick view of recent efforts, and they could look up concepts within resources using date as a cue without necessarily having to recall terminologies. Participants expressed that they could use granular time-based information to consider how well how they are able to pursue different goals within their day, week, or month. P10 shared how time-based overview could help self-learners plan their time better, especially if it offered them the flexibility to change the duration of the overviews: *"If I were to learn something over a month, I would first create weekly to-dos. After completing 4 weeks and before starting a new month, I would need a monthly picture [of things I have done]."*

Similarly, P15 added that a time-based overview over a larger span, such as several months, could also be helpful in assessing resources they *"have been visiting, but switched up later"*, and they could re-evaluate and reconsider their choices. P13 further shared how the Temporal Overview could reveal how effectively one engages with a resource. For example, *"visiting YouTube to watch one video but spending next half-hour watching cat videos"* would show up as a gap in learning and wasted time. P13 added that *"there isn't necessarily a correlation between time and what I learned"* without also considering, *"did I encounter a ton of problems and do I have to adjust my expectation when I can finish it?"*

Five other participants also voiced similar concerns about having a time-based overview as it could downplay the activities undertaken during a specific span of time and lead to ambiguity in interpretation (e.g., was the resource helpful or was it difficult?). Additionally, the granular hourly information could also become tedious to analyze for those who only wished to see a cumulative account of time (e.g., total time spent on a topic). Four participants mentioned that the time-based overview combined with or used in a sequence with resource-based overviews could better cater to their needs.

### 4.6.2 Helpful to reflect on resources and evolving objectives

Among the three probes, two were designed to highlight different aspects of resources used: while the Resource-based Overview showed a high level classification of the resource type, Topic-based Overview highlighted keywords and concepts from within collections of resources. We found that 5/12 participants ranked Resource-based as their first choice for overviews, while 7 others ranked it as their second preference, next to the Temporal Overview.

Half of the participants (6/12) preferred to first identify broad topics that they had learned about, followed by sub-categories. As the groups were distinct and labelled in the Resource-based Overview, participants found it easier to understand and interpret, and helpful for identifying the broad categories: *"I feel like it is important for me to know generally what topics I have been looking into, and their subtopics possibly." P11.* Additionally, participants' comments revealed that they wanted to have control over the topics appearing in the overviews. They expressed reluctance over automatically generated sub-topics in Topic-based Overview, stressing that an indicator of the fraction of content they had consumed or found relevant would be more useful. A summary of all the content within visited webpages which they may not have read entirely or found entirely

helpful would be less desirable. P18 said that learners needed to be more in control of identifying topics and their relevance, adding *"how [the topics] show up can be automated, but I want to decide what [content] is being shown [in the overview]."*

The majority of participants (9/12) wanted to focus their attention to specific parts of the overview to identify areas of weakness. Our use of the color red to represent items that required attention was particularly helpful for participants. As an improvement, five participants shared that *"it would be great to have a way to see outstanding questions"* (P13), short-notes to guide their attention to resources that need to be revisited. Participants added that resources that deal with the same category or subcategory of themes needed to be distinguishable from one another, such as, through the use of indicators of the required action (e.g., to read, to code, etc). Along the same lines, seven participants desired to see a better connection between learning objectives and resources to better plan their next session. In the context of project-driven learning, participants expressed a desire to see the relevance of resources with respect to the problems they were pursuing: *"I would look for a relation between the to-do list and what I have done [based on] the resources [in the overview]. It would be helpful to check if I am missing any items from my list."(P16)*

Although all of our participants found the idea of an integrated to-do list to be helpful, they said that feedback from mentors or peers would give them most clarity on their learning approach. Next, we describe how the participants speculated on using the different learning overviews to obtain feedback.

### 4.6.3 Desire to share learning histories for feedback

As previous research has shown, when learners rely on others to ask for help, they can struggle with describing their question or may not be able to articulate it using an accurate vocabulary [27]. We asked participants if they would willingly share the learning history overviews with others to seek feedback and the kind of information they would share or hide. All participants expressed that they would be strongly willing to share their learning histories with a more knowledgeable peer or mentor that they trusted and who could offer advice on their learning approach. They wanted a validation of the resources they had selected and be able to *"write down a question, then move on to learning something else, then come back to it or ask a colleague or an expert"* (P13).

Most participants (9/12) described different ways in which they would alter the overviews to make them more suitable for seeking feedback. For example, P09 shared how she would add annotations to the overview: *"I want to be able to send this [overview] to my peer and have them check out my bookmarks - I might add a note to let them know how the resource helped me."* P10 and P12 both mentioned that in technical subject areas, they would also add references to their implementation attempts, or any example questions they may have resolved: *"[For an applied skill] I would like to see a separate category [in Resource-based Overview] for practice sessions [arranged] by topics."* P18 added that while implementation could be an important detail to add for useful feedback, learners should have a choice to share it optionally, on request.

Some others (P16, P19, P20) also mentioned removing extra items from their learning histories. P20 shared how she was *"not comfortable sharing how [she] learned something, but if there was a way to filter out specific details [...] then that would be good."*. She preferred to abstract the information down to essentials, such as themes, titles, links, and sequence of visits. Overall, participants considered overviews to be helpful for *"show and tell"* and wanted to have editing controls to make them more suitable for feedback.

## 5 DISCUSSION

We have taken a learner-centered approach [63] to understand what matters to informal learners for self-monitoring and reflecting on

their learning experiences. Our work identifies key factors related to individual and social contexts of informal learners and how self-monitoring techniques could be adapted to address learner needs. We now reflect on the key insights from the elicitation studies and formulate the design requirements for self-monitoring interventions for informal learners.

## 5.1 Summary of Key Takeaways

Past studies have shown that informal learners often struggle to effectively self-direct their efforts, relying on suboptimal, trial-and-error approaches [9, 11, 21]. We explored different self-monitoring dimensions and visual overviews using paper-based mockups and interactive probes, taking inspiration from tracking tools used in productivity, health, and wellness. Our participants across both studies overall had a positive reaction to the idea of self-monitoring progress and the use of semi-automatic approaches, consistent with insights from other tracking domains [15]. Our key takeaways are:

1. **Interactive visual overviews could provide useful insight** for informal learners and create opportunities for self-reflections on various aspects of learning, such as, evaluation of the time utilized for different goal pursuits, or reconsideration of choices made in terms of resources and learning strategies.

2. **Semi-automated data-collection could encourage a learner** to be more engaged in self-monitoring behaviors. Automatic recording provides learners with a baseline of personal data which can then be refined with personally meaningful information, such as evolving goals, or nuances between resources.

3. **Overviews of learning data could facilitate reflections** through conversations with a trusted other when the learner has the controls to share and refine the presented data to add supplemental information, as well as remove data which is irrelevant for feedback purposes.

## 5.2 Key Requirements for Designing Self-Monitoring Interventions for Informal Learning

Based on the key findings from our studies, we synthesized key implications for the design of self-monitoring interventions for informal learners of computational skills.

### 5.2.1 Support automatic tracking and interactive overviews

The visual overviews in our study were considered helpful because they could serve as a quick at-a-glace summary [23] of learning efforts undertaken over a period through minimal learner intervention due to the automatic data-collection and generation of the overviews. However, to make visual overviews useful, any labelling achieved through automation should communicate how the labels were generated. Moreover, visualizations could be made more effective in drawing and guiding learners' attention by showing data in categories, such as broad topics or relevant resource meta-data, that learners can recognize. Learners should be supported in obtaining different kinds of insights through manipulation and interaction (e.g., select, filter, zoom) with the presented data [60].

### 5.2.2 Provide manual markers and personally relevant filters

To make the overviews personally relevant, learners should be offered semi-automatic support to mark the resources they have engaged with, such as through automatically suggested tags, along with the flexibility to specify their own tags, or marking up and filtering the resources according to metrics such as, perceived usefulness, level of difficulty, relevance to goals. Learners who are able to identify nuances between resources should be allowed to indicate the variations. For example, resource-specific indicators could help learners decide which resources to revisit, or show the extent to which learners have consumed a resource (or where they have left off), outstanding questions, or action-items (e.g., "to read", "to code", "to watch") associated with specific resources.

### 5.2.3 Reserve goal tracking for advanced stages of learning

While goal-tracking and checking for goal-alignment was unanimously considered a useful activity by all the participants in our study, there were apprehensions that such interventions may not be feasible for beginners. Learners may begin with broad goals, which may become more specific over time. Goal-tracking, therefore, should be adaptive, and suited to the stage of learning [42]. For example, learners could be given access to simple goal-trackers and reminders in the early stages of their learning. More sophisticated goal-tracking such as fine-tuned task identification, should be reserved for more advanced learners who may be at the application or implementation stage of their learning.

### 5.2.4 Allow tailoring of visual overviews for feedback

Self-monitoring can be made more effective with occasional feedback from an experts or knowledgeable peers [36]. We learned that visual overviews can be used to solicit feedback on learning strategies as they reveal the learner's pathways and attempts. However, our participants wished to tailor their overviews based on the type of feedback they wanted. This suggests that learners could benefit from the flexibility in determining the extent of detail to share for feedback. Interactive overviews should include ways to allow learners to add more context with further annotations, or remove details they consider redundant or irrelevant for the desired feedback.

### 5.2.5 Limitations and Future work

Although we recruited participants from different backgrounds and professions, there is a need for more studies with a more diverse set of participants of computational skills who may have different learning styles. Since the context we used in our scenarios was limited to online mediums only and our questions were focused on learning technical skills, whether our results will generalize to other informal learners should be further investigated. Moreover, the data we showed in the probes were curated by the researchers. More qualitative studies are required to uncover the nuances self-monitoring in domain-specific informal learning through the use of domain specific content, and ethical ways of using participants' own data. Future studies could also expand the informal learning context to physical resources and artifacts, and explore how physical tracking could be augmented by digital methods. Future studies could use experimental methods, observational methods, or in-situ data collection methods such as experience sampling, design probes or journaling, to triangulate the responses.

## 6 CONCLUSION

We have contributed insights on informal learners' perceptions of self-monitoring through an iterative design approach. Our synthesis reveals that learners in our study favored automatically generated interactive visual overviews of learning activities for its ability to facilitate awareness of their learning processes, but also expressed willingness to be involved with data collection for making them more suitable for reflections and planning. Our work opens up several opportunities for future research to use learner-centered approaches to understand and cater to informal and self-directed learners' needs.

## 7 ACKNOWLEDGMENTS

We thank the Natural Sciences and Engineering Research Council of Canada (NSERC) for funding this research.

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
