# OpenReview forum: "Exploring the Needs of Informal Learners of Computational Skills: Probe-Based Elicitation for the Design of Self-Monitoring Interventions"
_graphicsinterface.org/Graphics_Interface/2023/Conference — GI 2023_

### Official Review · Reviewer_G7Wx · 2022-12-30
**This paper presents two qualitative studies, using paper-mock-ups and interactive probes respectively, to investigate the design of a self-monitoring visualization tool for informal learners to track and reflect on their learning experiences of technical contents from a variety of resources and media. This is a very well written paper with good organizations.**

**Rating:** 9
**Confidence:** 4

**Review:**

The related work is well framed to motivate the current research. The two studies were well-designed with clear goals. The data collection and data analysis methods are both clearly described. The findings are also clearly presented with good insights. There are however a few minor issues that should be fixed/improved:
1.	Make sure all the figure references are correct (e.g., Design idea #2 on p.3) and figures are placed closer to where they are referenced (e.g., fig 1).
2.	Some of the findings in 4.3 are inconsistent. E.g., “Participants usually began by scanning for the overarching topics followed by sub-topics.” and “Half of the participants (6/12) preferred to first identify broad topics that they had learned about, followed by sub-categories. “ do not seem to be in alignment.
3.	5.1 and 5.2 have much overlaps. It’d improve succinctness by combining these two subsections.
4.	Since the contents used for the studies were technical, the paper title should reflect this narrower scope.

---

### Official Review · Reviewer_Znus · 2023-01-14
**The authors describe two elicitation studies to understand the design requirements of self-monitoring systems for informal learners. The authors provide details about the low and high fidelity views that the participants interacted with and provide useful guidelines for designing such systems. Overall, the paper is well written and the authors make important contributions to understand the design limitations and opportunities for self-monitoring systems for informal learners.**

**Rating:** 9
**Confidence:** 4

**Review:**

The authors motivate the need for this research by mentioning how informal learning has proliferated massively as the demand for developing skills on one’s own time has strongly increased. The authors design two elicitation studies with 8 and 12 participants respectively to gain insights into the kinds of information that informal learners would like to view in order to self-monitor, in addition to allowing useful interventions for helping them in the process.

Overall, the paper is well structured and motivated. The authors are tackling a big problem that will only increase in size as remote learning and the digital economy grow. The authors follow a principled approach of conducting an initial study with paper-based prototypes and then following it up with higher fidelity digital designs. The authors present a detailed analysis of the qualitative interviews conducted and draw themes from those interviews to succinctly present the findings. The authors make a strong conclusion to this paper by presenting four actionable design requirements that can be followed when designing future interfaces for informal learners.

There are some minor issues with the paper. The authors could have provided a more detailed explanation of the rationale behind choosing the three new summary designs in the interactive probes. In addition, it is unclear why the authors decided to recruit 12 new participants for the second study and not follow up with the original 8 to get their perspective on the new designs. There are also some minor typos and formatting issues - e.g. Section 3.1, Design Idea # 2 (fig. ??.2a).

---

### Official Review · Reviewer_ZaWP · 2023-01-16
**Nuanced needfinding studies with low-fi and interactive probes, but not persuasive enough for the applicability in the domains of learning**

**Rating:** 6
**Confidence:** 4

**Review:**

The paper presents two design requirement-elicitation studies with paper-based and interactive prototypes to explore ways to provide self-monitoring interventions for informal learners. Nuanced user preferences were elicited through interviews with a total of 20 participants in the two studies. The paper made the argument that the elicited preferences could lead to useful designs for the self-monitoring of learning.

Overall, I found the direction to design self-monitoring intervention for self-learning timely and important. The paper is well-motivated, nicely framed and quite well-written in narrating the problem space and the potential design dimensions involved. The paper is also deeply rooted in a user-centered design framework for eliciting requirements and building the interventions. The design approach is characterized by using early stage prototypes (can be interactive or not) to involve the target users since the beginning of a design process. The paper demonstrated applying the user-centered approach thoroughly with two rounds of probing studies. There were nuanced findings identified and summarized from the interviews.

With that said, the main concern I had is a methodological one, wondering whether a user-centered design approach is feasible for identifying true user needs and creating useful interventions for learning. Learning as complex cognitive and even social processes, it's unlikely learners are in the a good position to self-probe what's likely to work versus not for meaningful learning to happen. Another constraint is that both the lo-fi and hi-fi prototypes in the work are not functional to the extent to be used in real learning scenarios. The participants were not engaged as learners and could only report their perceptions, which are mostly first-impression preferences. It seems to me that these reports might be at best representing "user wants", and it's unclear whether they truly captured the "user needs".

Still I think this is an interesting work and can be of value to the audience of the community. I would recommend the authors to add reflections about the properties and constraints of the adopted method. A stronger connection with learning theories from a learning science perspective, as well as a clearer presentation of the contributions relative to the design and empirical literature can be helpful.

---

### Meta-Review · Area_Chair_JxG4 · 2023-01-18

**Recommendation:** 8
**Confidence:** 4

**Metareview:**

This paper describes two studies to elicit the requirements for interactive systems that can provide helpful interventions to assist informal learners self-monitor their progress.

Overall, I recommend acceptance for this paper. The reviewers have noted the timely nature of the topic and the growing need for self-monitoring as people learn new skills on their own time. The two studies were well-designed and the authors do a good job of extracting the themes in the qualitative interview data and presenting that to the reviewers. The authors make an important contribution by providing a thoughtful list of actionable design requirements.

The reviewers mentioned a few typos and inconsistencies in the figures that the authors can fix. In addition, (as one of the reviewers commented), it would be useful to add some brief information about the limitations of the adopted approach i.e. describing what the user "wants" and what the user "needs". Overall, these changes can be feasibly made in the camera ready version.